# Tribology and Anti-Ablation Properties of SiC-VN-MoS_2_/Ta Composite Coatings on Carbon/Carbon Composites from 25 to 800 °C

**DOI:** 10.3390/ma14226772

**Published:** 2021-11-10

**Authors:** Jun Cao, Jianbin Chen, Xinbo Wang, Jingbo Wen

**Affiliations:** 1School of Mechanical Engineering and Mechanics, Ningbo University, Ningbo 315211, China; chenjianbin@nbu.edu.cn; 2Ningbo HYRB Coating Co., Ltd., Ningbo 315220, China; 3Power Equipment Department, Shanghai Marine Diesel Engine Research Institute, Shanghai 201108, China; wxb2004929@126.com; 4Anhui Wuhu Meida Electromechanical Industry Co., Ltd., Wuhu 241199, China; mida@midabearing.com

**Keywords:** carbon/carbon composites, composite coatings, tribology, anti-ablation, wide range temperature

## Abstract

To improve the self-lubrication and anti-ablation performances of C/C (carbon/carbon) composites from 25 to 800 °C, we engineered three layers of composite coatings consisting of SiC–VN–MoS_2_/Ta to deposit on the surface of the C/C composites. The tribology and anti-ablation properties of the composite coatings were experimented under dry sliding wear. The equivalent stress and deformation of the composite coatings are studied. The results show that the CoFs (coefficients of friction) of the C/C composites are decreased by 156% at 800 °C due to the new generated self-lubricating compounds from the MoS_2_/Ta and VN coating. The anti-ablation of the C/C composites are improved by 25,300% due to the silicon glass, and the generated compounds from V, Mo and Si. The deformation of the C/C substrate under the protection of these coatings looks like a quadrangular star. The cack of the C/C composites is easily generated without the protection from coatings.

## 1. Introduction

C/C (carbon/carbon) composites consist of pyrolytic carbon and carbon fiber. The pyrolytic carbon is matrix, and the carbon fiber is the reinforcement. C/C composites are widely used in military and aerospace industries due to their low density, excellent thermal shock resistance and chemical stability [1,2]. However, properties such as tribology, oxidation and weight loss of the C/C composites are poor at high temperatures. It was found that the CoF (coefficient of frictional) of C/C composites from room temperature to 200 °C was less than 0.2 under the dry sliding wear, and it increased quickly to 0.4 at 650 °C [3,4,5]. The weight loss of the C/C composites increases with the temperature due to the oxidation and ablation of carbon. The weight loss of C/C composites was more than 5% at 600 °C, and more than 15% at 800 °C [6,7,8]. The C/C composites are mostly used as the materials of aircraft brakes, missile launcher nozzles and the mechanical parts of aerospace equipment at a wide range of temperatures. The self-lubricating and anti-ablation performances of C/C composites should be improved due to the poor performances at high temperature. There were many outstanding academic achievements of C/C composites. To improve the anti-ablation property of the C/C composites, the compounds of Si such as MoSi_2_, CrSi_2_ and SiC were studied. The oxygen was prevented due to the formation of liquid silicon glass at high temperature [9,10,11]. The ceramics such as ZrO_2_, SiC, Al_2_O_3_ and CrN were applied to improve the anti-wear and anti-oxidation of C/C composites [12,13,14,15]. However, the self-lubricating properties of the above ceramics were poor under dry sliding wear. The CoF of BN was more than 0.25 under a contact load of 10 N, and a dry sliding speed of 1 mm/s [16]. The CoF of SiC sintered with Al was higher than 0.6 under dry sliding wear [17]. The CoF of CrN was more than 0.6 under dry sliding wear [18]. The anti-ablation of C/C composites was improved through these functional coatings and materials. However, the self-lubricating properties of these ceramics were even worse than that of the C/C composites. In order to decrease the CoFs of C/C composites, the self-lubricating coatings such as graphite, PTFE and DLC (diamond-like carbon) were studied. However, their self-lubricating properties were invalid and poor at high temperature [19,20,21]. Though the CoFs of lubricants such as Ti_3_SiC_2_, Ti_2_AlC and CaF_2_/BaF_2_ were lower than that of C/C composites at 800 °C, their lubricating properties were poor at room temperature [22,23]. For example, the CoF of Ti_2_AlC was less than 0.3 at 800 °C under dry sliding wear. However, it was more than 0.5 at room temperature [24]. Recently, rare earth materials have been used in composite coatings. The tribological properties of composite coatings improved as the materials of LaF_3_, CeO_2_ and Y_2_O_3_ were applied [25,26,27]. The anti-ablation performance of C/C composites has improved. However, the self-lubricating properties of the C/C composites are still poor. The coatings and materials not only should have low CoFs but also excellent anti-oxidation properties.

The tribology and anti-ablation of C/C composites should be improved at high temperatures. It was found that the compounds of V, Mo and Ta have excellent self-lubricating and high anti-ablation ability at high temperatures. In this study, we designed three layers of composite coatings consist of the SiC-VN-MoS_2_/Ta and deposited them onto the surface of a kind of C/C composites. The tribology and anti-ablation of the C/C composites are hypothesized to be improved by the new generation compounds from the SiC–VN–MoS_2_/Ta. We experimented the tribology and anti-ablation properties of the composite coatings at room temperature and 800 °C under dry sliding wear, and studied the equivalent stress and deformation of composite coatings under normal pressure. The mechanisms of tribology and anti-ablation are discussed through these experiments and results.

## 2. Experiment

### 2.1. Preparations of the Composite Coatings

The density of C/C composites is 1.91 g/cm^3^. The sizes of the tested specimens are shown in Figure 1. Firstly, all samples were polished by 400# and 600# SiC paper successively. The surface roughness of Ra of all samples is 0.85 ± 0.03 μm. The samples were ultrasonic cleaned under acetone liquid for 10 min to remove the impurities. At last, all samples were dried at 80 °C for 30 min.

The bonding strength of coatings on the surface of the C/C composites is influenced by CTE (coefficient of thermal expansion). The bonding strength will be decreased if the difference of the CTE increases [28]. The CTE of C/C composites is 1.5 × 10^−6^/k, which is less than that of most materials. For example, the CTE of B_4_C is 6 × 10^−6^/k [29]. The CTE of MoSi_2_ and MoS_2_ is 8.1 × 10^−6^ and 10.8 × 10^−6^/k, respectively [30,31]. The CTE of SiC is 2.4 × 10^−6^/k, which is close to that of C/C composites. The anti-oxidation of the C/C composites will be improved if a SiC coating is applied on the surface of the C/C composites. The oxygen was prevented as the liquid silica glass distributed on the surface of the C/C composites at high temperature [32,33]. The shortage of SiC was the high CoF at room temperature and high temperature [34,35]. In this study, the SiC coating was deposited as a transition layer to connect with the self-lubricating materials with higher CET than that of the C/C composites.

The thermal spraying technology is not suitable for the deposition of SiC on the surface of the C/C composites. SiC will be gasified at the thermal spraying temperature, which is higher than 2700 °C. The SiC coating is hardly deposited by cold spraying due to the poor plastic deformation. The best way to deposit SiC coating is the pack cementation under the chemical reaction of Si and graphite. The Si, C and Al_2_O_3_ powders are used to prepare the SiC coating on the surface of the C/C composites. The purity of Si, C and Al_2_O_3_ powers are more than 99.98%, and their average sizes are 30 ± 3 μm. The Al_2_O_3_ powers are used to improve the adhesion strength of SiC coating on the surface of the C/C composites, due to the large specific surface area of the melted Al_2_O_3_. The SiC coating was prepared as follows. Firstly, these samples of the C/C composites were buried and surrounded by these three mixed powders in a graphite crucible. Then, they were heated in a stove at 1350 °C under a normal pressure of Ar gas for 1.5 h. At last, all samples were cleaned in distilled water under ultrasonic vibration to remove the residue powders.

The oxides of VN such as V_2_O_5_ and V_4_O_9_ were the self-lubricating materials with low CoFs at high temperature [36,37]. In addition, the CoF was decreased in some reports as the compounds generated from VN with Ag [38]. Based on the assumption that the CoF of the compounds of VN and Ta will be decreased at high temperature, the VN and Ta were applied. The purity of the VN powers was more than 99.98%, and their average sizes were 40 ± 3 μm. The VN coating was prepared on the surface of the SiC coating. Then, the top coating consisted of Ta that was deposited on the surface of the VN coating. The VN coating was prepared by plasma spraying. The parameters of the coating deposition are listed as Table 1.

The CoF of VN at room temperature was more than 0.4 [39]. We found that the CoF of MoS_2_ was low at room temperature. However, the anti-wear and CoF were poor at high temperature. MoS_2_ loses efficacy at high temperature, because it will be oxidized and decomposed. We found that the CoFs of the compounds such as Ag_2_MoO_4_, BaMoO_4_ and Ag_3_VO_4_ were lower than those of VN and MoS_2_ at high temperatures [38,40]. Thus, the tribological property of C/C composites will be improved from room temperature to 800 °C under the coating consisting of Ta and MoS_2_.

Based on the assumption that the CoF of compounds generated from MoS_2_, VN and Ta will be low at high temperature, three layers of C/C composites were designed in this study. From the bottom to the top, the sandwich layers deposited on the surface of C/C composites were the SiC, VN and MoS_2_/Ta. The layer of MoS_2_/Ta was deposited by liquid spraying. Ta powder was 100 nm, and its purity was 99.98%. MoS_2_ powder was 100 nm, and its purity was 99.99%. The mixed ratios of Ta and MoS_2_ powders were 25% and 75%, respectively. Ta and MoS_2_ powders were not deposited by thermal spray, because these powders are oxidized at high temperature from the thermal spray flame. Liquid spraying was the deposition method of coating formation. The mixed liquids consisted of PI (Polyimide), epoxy resin of E-44, acetone and the mixed powders. The PI and the epoxy resin were used as binders, and the acetone was a solvent. The preparation details of top layer are presented as follows: the spraying distance was 80 mm; moving speed of gun was 100 mm/s; and spraying angle was 95° ± 3°. After spraying, all samples were heated under 220 °C in the Ar gas protection stove for 2 h. The three layers of composite coatings are shown in Figure 2.

### 2.2. Frictional Experiments at Room Temperature and 800 °C

All samples were tested using a ball-on-disk tribometer of UTM-2 friction and wear test equipment (Center for Tribology (CETR), Campbell, CA, USA). All samples were tested three times to eliminate the randomness of experiments. The normal contact load was 5 N, which is the same load as a kind of roll bearing made from C/C composites [41]. The working temperature increased from room temperature to 800 °C. Thus, two temperatures of 25 and 800 °C were applied in this study. The material of the ball was the 440 C stainless steel. The roughness of Ra of the contact surface was 0.8 μm. The experiment details are listed in Table 2. The CoFs of the C/C composites and the coatings under different temperatures are shown in Figure 3.

### 2.3. Weight Loss at 800 °C

The high CoF and oxidation at high temperature are the two shortages of the C/C composites. We found that the composite coatings of the SiC-VN-MoS_2_/Ta have high-efficiency to decrease CoF at 800 °C. The C/C composites are easily oxidized at high temperature. The C will be decomposed as the CO_2_ and CO. The formula of weight loss of oxidation is expressed as follows [42]:(1)$$w=\frac{m_{0}-m_{1}}{m_{0}}\times 100\%$$
where *w* is the weight loss, $m_{0}$ is the mass before the test and $m_{1}$ is the mass after the test.

The ablation tests accompanied the frictional experiments in the UTM-2 friction and wear test equipment. The temperature first increased from room temperature to 800 °C. Then, for the ablation test, the temperature remained at 800 °C for 15 min. At last, the temperature decreased from 800 °C to room temperature. The weight loss of the C/C composites under the protection of three layers composite coatings is shown in Figure 4.

### 2.4. Bearing Capacity and Crack Generation Analysis

The bearing capacity and crack generation of coatings are not clear and hard to measure during the frictional movements at different temperatures. The equivalent stress and the deformation of the composite coatings are studied in this paper to illustrate the bearing capacity and the crack generation. The studies of stress and deformation of composite coatings were simulated by ABAQUS. Before the simulation, the modulus of elasticity and Poisson’s ratio of the MoS_2_/Ta composite coatings were experimented by the tensile testing machine of MTS and nano indentation instrument of Hysitron, respectively. The mechanical properties of the composite coatings of the SiC-VN-MoS_2_/Ta are shown in Table 3 [43,44].

The bearing capacity and crack generation of the composite coatings were studied at room temperature. The properties at 800 °C have not been studied for the following reasons. Firstly, it is very hard to measure the physical properties of these composite coatings at 800 °C. Secondly, the analysis method can be referenced from the results at room temperature in this paper. Thus, only bearing capacity and the crack generation under room temperature are shown in this paper. The method and the results presented here can be referenced to study the bearing capacity and crack generation if the physical properties of these sandwich coatings are measured at 800 °C in the future. The explicit dynamic simulation was applied for this study. The contact surface of the frictional ball and the surface of the MoS_2_/Ta coating is defined as frictional contact under the CoF of 0.22. The surfaces of the VN, SiC and C/C composites are all defined as bonded contacts which cannot separate from each other. Based on the relative motion, the bottom surface of the C/C base was fixed, and the frictional ball was restrained as a rigid body moving at a speed of 10 mm/s parallel to the coating surface. During the simulation, a 5 N concentrated force was applied on the reference point located at the center of the frictional ball. During the solving of the surface-to-surface contact problem, the mechanical constraint formulation and sliding formulation were set as kinematic contact method and small sliding, respectively. These coatings were regarded as isotropy during the finite element analysis [45,46]. The meshes of these three coatings are defined as the eight-node hexahedral element. The contact surfaces are finely meshed. The details of the meshes of all parts are shown in Figure 5.

By the finite element analysis, the results of the bearing capacity and the crack generation of the composite coatings are shown in Figure 6.

To analyze the equivalent stress and the deformation of the C/C substrate under protection of the composite coatings, the comparisons of the C/C substrate without the coatings are shown in Figure 7.

## 3. Results and Discussions

### 3.1. Tribological Properties at Different Temperature

The three layers of composite coatings are shown in Figure 2. We found that the thickness of the SiC coating was 5 ± 2.5 μm. The thicknesses of VN coating and MoS_2_/Ta coating were 20 ± 2 μm and 12 ± 2 μm, respectively. These coatings were compact. There were no cracks among different layer interfaces. The CoFs of C/C substrate at room temperature and at 800 °C are shown in Figure 3. The average CoFs of the composite coatings at room temperature and 800 °C were 0.22 and 0.28, respectively. The average CoFs of the C/C substrate at room temperature and 800 °C were 0.16 and 0.46, respectively. Although the lubrication performance of the composite coatings was worse than that of the C/C composites, the anti-wear and the wear depth of the composite coating improved. The wear marks were scanned by the white light interference 3D profilometer of Contour Elite I (Bruker, Madison, WI, USA). As shown in Figure 8, the wear depth and width of the composite coatings are smaller than that of the C/C substrate. As the frictional ball moved on the surface of C/C composites, the carbon was crushed. These tiny wear details can be ground to transfer film. Thus, the C/C composites achieved low CoF, and had self-lubrication performance. However, the anti-wear of the pyrolytic carbon and carbon fiber is poor. The C/C substrate is easily sheared. As these three layers of composite coatings were applied on the surface of the C/C substrate, the anti-wear property improved due to the hardness of the SiC, VN and Ta. The CoF of the PI, SiC and Ta are higher than that of carbon at room temperature. This is why the CoF of the C/C substrate was lower than that of the composite coatings at room temperature.

The CoF of C/C substrate significantly increased to 0.46 under the temperature of 800 °C. The high CoF is due to the chemisorption of oxygen on the surface of C/C composites [4]. There was no self-lubricating performance, because C/C composites are oxidized at high temperature. The carbon of the C/C substrate loses lubrication. It was burnt, and the surface of C/C substrate was rough due to the ashes. The CoF of composite coatings was 0.28 under the temperature of 800 °C. The low CoF at high temperature is attributed to the new generations of Ta*_x_*Mo*_y_*O*_z_*, Ta*_x_*V*_y_*O*_z_*, V*_x_*O*_y_* and Mo*_x_*O*_y_*. The CoFs decreased due to these self-lubricating materials generated at high temperature.

### 3.2. Anti-Ablation Properties at 800 °C

The weight loss of C/C composites was 22.86%, as shown in Figure 4. However, the weight loss of C/C substrate was only 0.09% under the protection of multiple coatings. The wight loss of the C/C composites was higher than 20% and even 100% at high temperature. The CO and CO_2_ will be generated and escape from the C/C composites. The weight loss of the C/C aircraft brake was 1.18% at 700 °C as the B_4_C and borosilicate glass were applied on their surface [47]. The weight loss of the C/C composites was 0.41% at 700 °C under the protection coating consisting of B_2_O_3_ and SiO_2_ [48]. The reasons are presented as follows. The solid SiC melts at 800 °C. The C/C substrate is enclosed with liquid glass, and oxygen is prevented from contacting carbon. The CO and CO_2_ are prevented from the C/C composites. Additionally, the oxygen is absorbed from the air, and reacts with SiC, V and Mo at high temperature. The weight is increased by the new generated SiO_2_ and V*_x_*O*_y_* at high temperature. Thus, the wight loss of C/C composites is only 0.09% under the protection from SiC–VN–MoS_2_/Ta.

### 3.3. Bearing Capacity and Crack Generation

The stresses and deformations of the composite coatings are shown in Figure 6 and Figure 7. The equivalent stresses of the MoS_2_/Ta coating, VN coating and the SiC coating are 27.03 MPa, 13.62 MPa and 12.02 MPa, respectively. The nephograms of these stresses are not symmetrical. The stress of the MoS_2_/Ta coating distributes like a narrow ellipse. However, the regularity and symmetry of the ellipse shown in Figure 6b,c are better than that of the stress distribution shown in Figure 6a. As the frictional ball moves forward, the stress distribution varies with the direction of the frictional ball. Firstly, the VN coating and the SiC coating do not contact the frictional ball directly. Secondly, the physical properties of these coatings are different. Thus, the stress distributions of the VN coating and the SiC coating are different from that of the MoS_2_/Ta coating. The deformations of the MoS_2_/Ta coating, VN coating and SiC coating at the normal direction are 0.2751, 0.1750 and 0.1283 μm, respectively. There are many wrinkles presented on the surface of the MoS_2_/Ta coating. The deformations of the VN coating and the SiC coating look like five-pointed star and quadrangular star, respectively. The contact style of the frictional pair is the point to surface. A small circle point appears in all pictures of Figure 6 and Figure 7. The top layer coating is sheared by the reciprocating frictional ball. These wrinkles and cracks are generated due to the large normal force and shear force from the moving ball. However, the VN coating and SiC coating mainly bear the normal force. The tiny shear force is just from the local deformation of the MoS_2_/Ta coating. The coating collapses around the contact point under the normal force. Thus, the deformation looks like a star. Moreover, these three coatings have support actions. The deformation of the C/C substrate shown in Figure 7b is better than that of the SiC coating shown in Figure 6f, and the deformation of the SiC coating is better than that of the VN coating shown in Figure 6e. The farther away the frictional ball is, the smaller the crack generation will be. This result also can be found in Figure 7b,d. Without the protection of composite coatings, the crack generation of C/C composites is easy. The numerical values of the stress distribution and deformation of the C/C composites are much higher than that of C/C substrate under the coating protection. This illustrates that the bearing capacity of C/C substrate is improved by these composite coatings. There are some cracks presented in Figure 7d. The deformation of C/C composites is much worse than that of MoS_2_/Ta coating under the same contact. However, the deformation of the C/C substrate under the coating protection looks like a quadrangular star.

The tribology and weight loss are serious problems of aircraft brakes, nozzle throats of rocket engines and bearings made by C/C composites at high temperatures. Though the weight loss of C/C composites could be sharply reduced by orders of magnitude under the protection of composite coatings such as B_4_C, ZrB and CrSi_2_, the self-lubrication was still poor at high temperature. However, the new designed composite coatings of SiC–VN–MoS_2_/Ta have the excellent properties of tribology and anti-ablation from room temperature to 800 °C.

## 4. Conclusions

Three layers of composite coatings consisting of SiC, VN and MoS_2_/Ta were designed and experimented at a wide range of temperatures. Through these results of dry sliding wear and ablation, the following conclusions are drawn.

(1)The CoF of C/C composites decreased by 156% when the SiC–VN–MoS_2_/Ta were coatings prepared at 800 °C. The oxides and compounds of V, Mo and Ta have self-lubricating performances at high temperatures.(2)The weight loss of C/C composites at 800 °C decreased by 25,300% under the protection of SiC–VN–MoS_2_/Ta coatings. Oxygen is prevented from contacting C/C composites.(3)The deformation of the C/C substrate under the coating protection looks like a quadrangular star. Without coating protection, cracks appear on the surface of the C/C composites. The cack of the C/C composites is easily generated without the coating protection.

## Figures and Tables

**Figure 1 materials-14-06772-f001:**
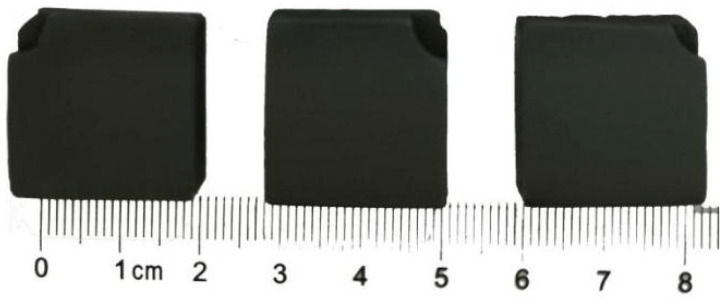
Tested specimens of C/C composites.

**Figure 2 materials-14-06772-f002:**
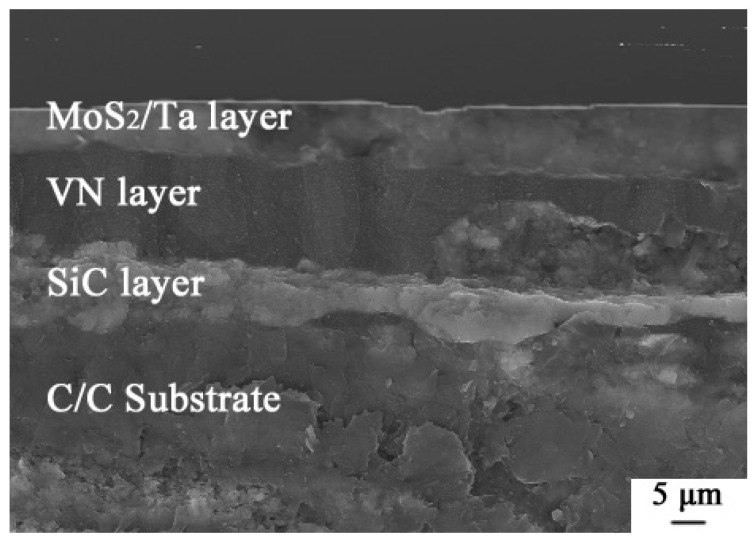
Three layers of composite coatings.

**Figure 3 materials-14-06772-f003:**
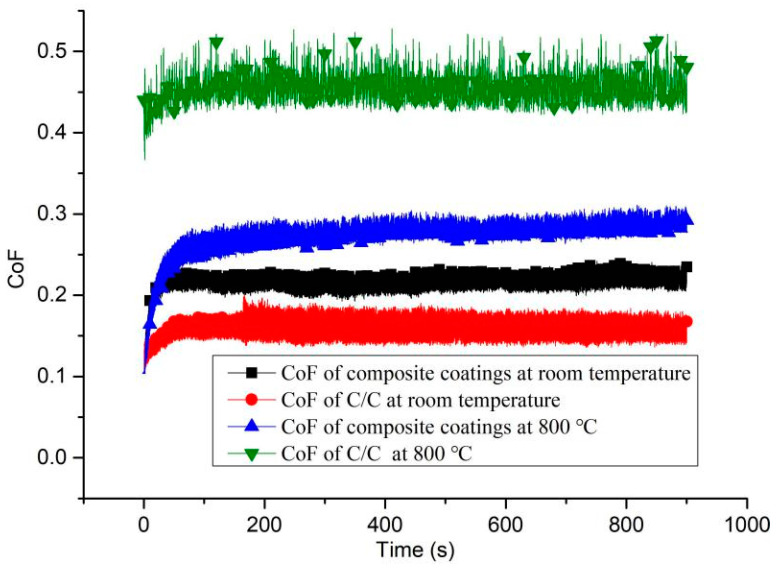
CoFs of the C/C composites and coatings under different temperatures.

**Figure 4 materials-14-06772-f004:**
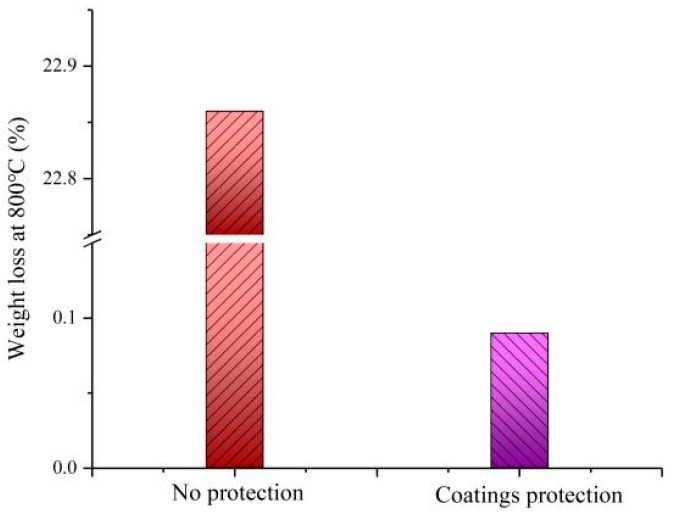
Weight loss of the C/C composites at 800 °C.

**Figure 5 materials-14-06772-f005:**
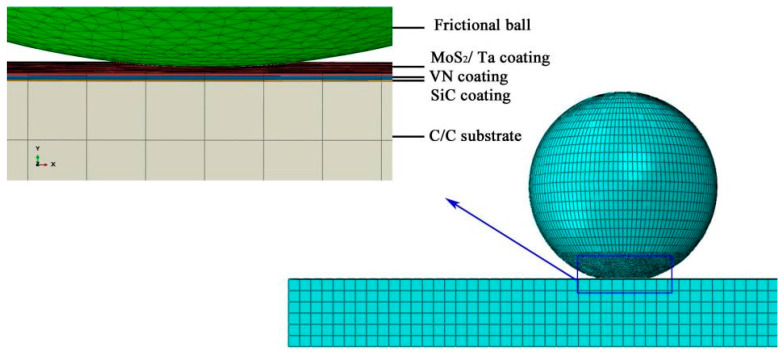
Mesh model of the frictional pairs.

**Figure 6 materials-14-06772-f006:**
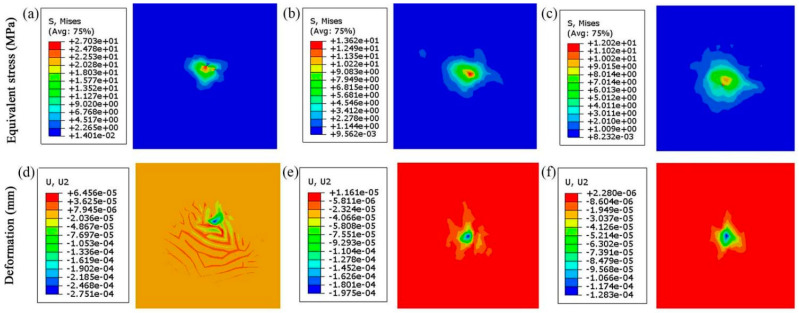
Stress and deformation of three coatings: (**a**) stress of MoS_2_/Ta coating, (**b**) stress of VN coating, (**c**) stress of SiC coating, (**d**) deformation of MoS_2_/Ta coating, (**e**) deformation of VN coating, and (**f**) deformation of SiC coating.

**Figure 7 materials-14-06772-f007:**
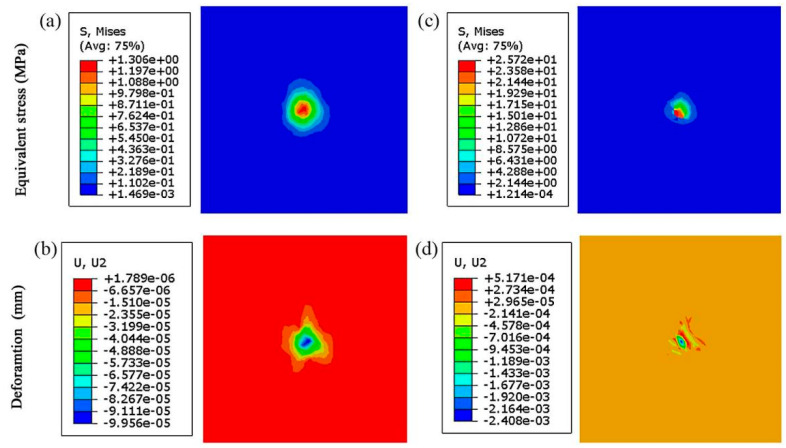
Stress and deformation of the C/C composites: (**a**) stress of the C/C composites without coatings, (**b**) deformation of the C/C composites without coatings, (**c**) stress of C/C composites with coatings, and (**d**) deformation of C/C composites with coatings.

**Figure 8 materials-14-06772-f008:**
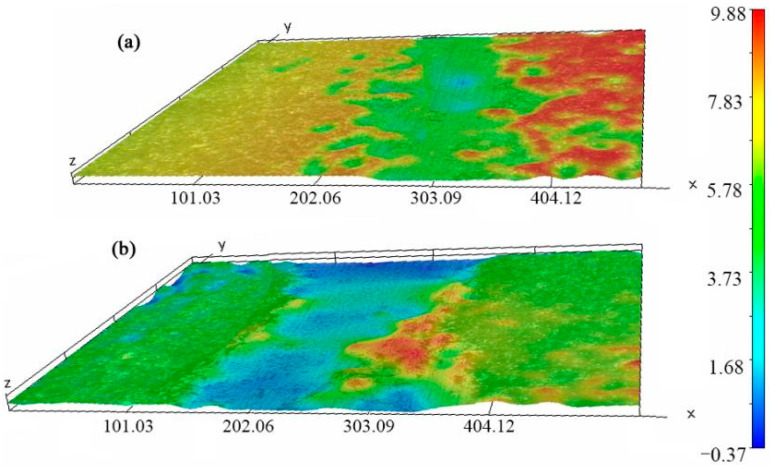
Wear depth and width of the composite coatings and the C/C composites. (**a**) C/C composites with coatings; (**b**) C/C composites without coatings.

**Table 1 materials-14-06772-t001:** Preparation details of the VN coating.

Ar	H_2_	Spraying Distance	Powder Feed Rate	Spraying Power
12 slpm	10 slpm	100 mm	15 g/min	38 KW

**Table 2 materials-14-06772-t002:** Details of frictional test.

Work Condition	Frictional Distance	Contact Load	Sliding Velocity	Time	Temperature
	mm	N	mm/s	min	°C
Dry sliding wear	5	5	20	30	25/800

**Table 3 materials-14-06772-t003:** Mechanical properties of the SiC-VN-MoS_2_/Ta.

Mechanical Property	SiC	VN	MoS_2_/Ta	C/C	Ball
Modulus of elasticity (GPa)	410	178	27.57	21	211
Poisson’s ratio	0.14	0.29	0.37	0.25	0.30
Density (kg/m^3^)	3100	6130	4560	1910	7850

## Data Availability

Not applicable.

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
