# Peer review of "Tribology and Anti-Ablation Properties of SiC-VN-MoS2/Ta Composite Coatings on Carbon/Carbon Composites from 25 to 800 °C"

_materials, 2021, doi:10.3390/ma14226772_

Round 1

Reviewer 1 Report

This article concerns the analysis of the influence of temperature ranging from 25 to 800 for composite coatings on Carbon / Carbon on tribological and anti-ablation properties. I assess the structure and division of the content as correct.

The main problem in this chapter is formulation and numerical analysis. It is at an insufficient level in terms of providing basic information on the formulation, i.e. boundary conditions, and how the contact problem is solved using the finite element method.

Moreover, the main problem of this task is that the contact problem is a nonlinear problem and requires very careful numerical calculations.

The finite element mesh shown in Fig. 4 and the results in Figs. 5 and 6 do not confirm the correctness of the performed calculations. This chapter requires a complete rethinking and does not allow me to form a positive opinion.

Author Response

Responses to the comments

Dear editors and referees,

We are grateful for the positive and helpful comments by reviewers. Based on the comments we have papered a revised version of the manuscript which will hopefully clarify the remaining questions. Words are highlighted with red in the revised manuscript. Please find our details answers. Thank you very much!

In this section, the answers to the comments are marked with red color.

1. This article concerns the analysis of the influence of temperature ranging from 25 to 800 for composite coatings on Carbon / Carbon on tribological and anti-ablation properties. I assess the structure and division of the content as correct.

The main problem in this chapter is formulation and numerical analysis. It is at an insufficient level in terms of providing basic information on the formulation, i.e. boundary conditions, and how the contact problem is solved using the finite element method.

Moreover, the main problem of this task is that the contact problem is a nonlinear problem and requires very careful numerical calculations.

The more details of the simulation are presented as follows.

The explicit dynamic simulation is applied for this study. The contact surface of the frictional ball and the surface of the MoS2/Ta coating is defined as frictional contact under the CoF of 0.22. The surfaces of the VN, SiC and C/C composites are all defined as bonded contacts which can not separate with each other. Based on the relative motion, the bottom surface of the C/C base is fixed and the frictional ball is restrained as rigid body moving at a speed of 10 mm/s parallel to the coating surface. During the simulation, a 5 N concentrated force is applied on the reference point locating at the center of the frictional ball. During the solving of the surface-to-surface contact problem, the mechanical constraint formulation and sliding formulation are set as kinematic contact method and small sliding, respectively. These coatings are regarded as isotropy during the finite element analysis [45, 46]. The meshes of these three coatings are defined as the eight-node hexahedral element. The contact surfaces of are finely meshed.

[45] Marcinowski, J.; Różycki, Z.; Sakharov, V. Numerical simulations of destructive tests of cast iron columns strengthened with a CFRP coating. Materials. 2020, 13,11-18.

[46] Wang, K.S.; Zhang, Fa.Z.; Bordia, R.K. FEM modeling of in-plane stress distribution in thick brittle coatings/films on ductile substrates subjected to tensile stress to determine interfacial strength. Materials. 2018, 11,497.

2. The finite element mesh shown in Fig. 4 and the results in Figs. 5 and 6 do not confirm the correctness of the performed calculations. This chapter requires a complete rethinking and does not allow me to form a positive opinion.

The finite element mesh had been improved. As shown in Fig. 4, the meshes of contact surfaces are refined. In this section, the stress and deformation of these coatings and the C/C substrate are studied to analyze the bearing capacity and the wear marks. The details and the processes of the simulations are checked to ensure the correctness of these results.

3. In introduction section, more recent and relevant papers may be cited related to this topic.

The references published in recent three years are cited to replace the original relevant papers.

Reviewer 2 Report

The authors have conducted a good experimental work to find out Tribology and Anti-ablation Properties of SiC-VN-MoS2/Ta 2 Composite Coatings on Carbon/Carbon Composites from 25 ℃ 3 to 800 ℃. But certain areas need improvement. The authors may like to address following comments:

     1. The introduction section lacks the citation of recent papers to warrant the claimed research gap. 

     2. Avoid long paragraphs, e.g. lines 25-61.
     3. Avoid small paragraphs, e.g. lines 199-201
     4. Heading 3 should have subheadings to explain different aspects of
outcomes.
     5. Stress and deformation areas should be labelled in figures 5 and 6.
     6. In current form, results have been explained with mere observations. There is a need of elaboration with scientific reasoning.
     7. It is encouraged to explain the possible implementation of this research in real life applications.

     8. Reported 25300% should be revisited and reconfirmed.
     9. English language should be improved throughout the manuscript.
The whole article needs thorough proofread by an English expert.

Author Response

Responses to the comments

Dear editors and referees,

We are grateful for the positive and helpful comments by reviewers. Based on the comments we have papered a revised version of the manuscript which will hopefully clarify the remaining questions. Words are highlighted with red in the revised manuscript. Please find our details answers. Thank you very much!

In this section, the answers to the comments are marked with red color.

1. Avoid long paragraphs, e.g. lines 25-61.

The English has been improved.

2. Avoid small paragraphs, e.g. lines 199-201.

The English has been improved.

3. Heading 3 should have subheadings to explain different aspects of outcomes.

The subheadings of results and discussions have been added.

4. Tested specimens should be shown with labelling.

The tested specimens shown with label are presented.

5. Results should be more elaborated with scientific reasoning. In current form, it looks like a lab report.

More details and reasons have been discussed in the section of results. The revised and added details marked with red color in the revised paper.

6. A brief section should be added before conclusion section about the implementation of this research in real field for practicing professionals.

A brief section about the implementation of this research in real field has been added. 

The tribology and weight loss are the serious problems of the aircraft brake, nozzle throat of rocket engine and bearing made by C/C composites at high temperature. Though the weight loss of C/C composites could be sharply reduced by orders of magnitude under the protection of composite coatings such as B4C, ZrB andCrSi2, the self-lubrication was still poor at high temperature. However, the new designed composite coatings of SiC-VN-MoS2/Ta have the excellent properties of tribology and anti-ablation from room temperature to 800 ℃.

7. Reported 25300% should be revisited and reconfirmed.

The data has been confirmed. The more details have been discussed.

The wight loss of the C/C composites was higher than 20% and even 100% at high temperature. The CO and CO2 will be generated and escape from the C/C composites. The weight loss of the C/C aircraft brake was 1.18% at 700 ℃ as the B4C and borosilicate glass were applied on the surface of the C/C composites [47]. The weight loss of the C/C composites was 0.41% at 700 ℃ under the protection coating consisting of B2O3 and SiO2 [48]. The reasons are presented as follows. The solid SiC will be melt at 800 ℃. The liquid glass encloses the surface of C/C substrate, which prevents the oxygen to contact with the carbon. The CO and CO2 are prevented from the C/C composites. What is more, the oxygen is absorbed from the air, and reacts with SiC, V and Mo at high temperature. The weight is increased by the new generated SiO2, VxOy at high temperature. Thus, the wight loss of C/C composites is only 0.09% under the protection from SiC-VN-MoS2/Ta.

8. English language should be improved throughout the manuscript. The whole article needs thorough proofread by an English expert.

The English has been improved.

Round 2

Reviewer 1 Report

In the presented work, the authors corrected most of the reviewer's comments. However, one question still remains unanswered. Solving contact issues requires a very careful approach and experience in FEM calculations. The authors improved the model and the obtained results differ significantly from those presented in the previous version. There is some concern that this was left without comment in the response. The problem that remains to be solved is the depicted generation of cracks. The authors ignore this fact. In such a complex model, the way in which cracks are considered and how they affect the contact surface is very interesting. This can be seen in the presented figures 6 and 7. I would like to ask you to present a broader analysis of this problem, because it is very interesting, especially from the point of view of the cracking mechanics of composite coatings.

Author Response

Responses to the comments

Dear editors and referees,

We are grateful for the positive and helpful comments by reviewers. Based on the comments we have papered a revised version of the manuscript which will hopefully clarify the remaining questions. Words are highlighted with red in the revised manuscript. Please find our details answers. Thank you very much!

In this section, the answers to the comments are marked with red color.

In the presented work, the authors corrected most of the reviewer's comments. However, one question still remains unanswered. Solving contact issues requires a very careful approach and experience in FEM calculations. The authors improved the model and the obtained results differ significantly from those presented in the previous version. There is some concern that this was left without comment in the response. The problem that remains to be solved is the depicted generation of cracks. The authors ignore this fact. In such a complex model, the way in which cracks are considered and how they affect the contact surface is very interesting. This can be seen in the presented figures 6 and 7. I would like to ask you to present a broader analysis of this problem, because it is very interesting, especially from the point of view of the cracking mechanics of composite coatings.

The more discussions are presented as follows.

The deformation of the MoS2/Ta coating, VN coating and the SiC coating at the normal direction is 0.2751 μm, 0.1750 μm and 0.1283 μm, respectively. There are many wrinkles presented on the surface of the MoS2/Ta coating. The deformation of VN coating and the SiC coating looks like five-pointed star and quadrangular star, respectively. The contact style of the frictional pair is the point to surface. It is found that a small circle point appears in all pictures of Fig. 6 and Fig. 7. The top layer coating is sheared by the reciprocating frictional ball. These wrinkles and cracks are generated due to the big normal force and shear force from the moving ball. However, the VN coating and SiC coating bear the normal force mainly. The tiny shear force is just from the local deformation of the MoS2/Ta coating. The coating collapses around the contact point under the normal force. Thus, the deformation looks like a star. What is more, these three coatings have support actions. The deformation of the C/C substrate shown in Fig. 7(b) is better than that of the SiC coating shown in Fig. 6(f), and the deformation of the SiC coating is better than that of the VN coating shown in Fig. 6(e). The farther away the frictional ball is, the little the crack generation will be. This result also can be found in the Fig. 7(b) and Fig. 7(d). Without the protection of composite coatings, the crack generation of C/C composites is easy.

Reviewer 2 Report

The paper is well revised.

Round 3

Reviewer 1 Report

The authors took into account all comments of the reviewer.